# Study on Pre-Exposure Prophylaxis Regimens among Men Who Have Sex with Men: A Prospective Cohort Study

**DOI:** 10.3390/ijerph16244996

**Published:** 2019-12-09

**Authors:** Dan Wu, Hao Tao, Jianghong Dai, Hao Liang, Ailong Huang, Xiaoni Zhong

**Affiliations:** 1Department of Epidemiology and Health Statistics, School of Public Health and Management, Chongqing Medical University, Chongqing 400016, China; iron_wd999@163.com (D.W.); taohao_cqmu@163.com (H.T.); 2Research Center for Medicine and Social Development, Chongqing Medical University, Chongqing 400016, China; 3Innovation Center for Social Risk Governance in Health, Chongqing Medical University, Chongqing 400016, China; 4Department of Epidemiology and Health Statistics, School of Public Health, Xinjiang Medical University, Xinjiang 830011, China; epi102@sina.com; 5Department of Epidemiology and Health Statistics, School of Public Health, Guangxi Medical University, Nanning 520021, China; haolphd@163.com; 6Key Laboratory of Molecular Biology, Ministry of Molecular Biology, Infectious Diseases, Chongqing Medical University, Chongqing 400016, China; ahuang1964@163.com

**Keywords:** MSM, PrEP, regimens, adherence

## Abstract

*Background*: There are limited studies on the medication regimen of pre-exposure prophylaxis (PrEP) among men who have sex with men (MSM) in China. This study compared the effectiveness of and adherence to two prophylactic HIV medication regimens, which provided evidence and guidance for the application and promotion of the PrEP strategy in the MSM population in China. *Methods*: We conducted an open, non-randomized, multicenter, parallel, controlled clinical intervention study in western China. Subjects were recruited by convenience sampling at research centers in Chongqing, Guangxi, Xinjiang and Sichuan, China from April 2013 to March 2015, and they were categorized into the daily PrEP, event-driven and blank control groups. Tenofovir disoproxil fumarate (TDF; 300 mg/dose) was administered to subjects in the daily PrEP and event-driven groups, and all subjects were followed up every 12 weeks for 96 weeks. Demographic, behavioral, psychological characteristics and AIDS-related attitudes were assessed using self-completed questionnaires. TDF serum concentrations in subjects in Chongqing and Sichuan were quantified after systematic sampling. *Results*: Of the 2422 enrolled MSM, 856 were eligible for statistical analysis (PrEP group: 385 and event-driven group: 471); 30 and 32 subjects in daily PrEP and event-driven groups, respectively, were HIV-positive; the incidence of HIV infection was as follows: daily PrEP group, 6.60 cases/100 person-years and event-driven group, 5.57 cases/100 person-years (relative risk (RR) 95% confidence interval (CI) was 0.844 (0.492–1.449)); HIV incidence did not differ significantly when stratified by medication adherence or sites. When the medication adherence rate was ≥80%, the median TDF serum concentrations were 0.458 mg/L, and 0.429 mg/L in the daily PrEP, and event-driven groups, respectively (not significant; *p* > 0.05); Subjects who were in the event-driven PrEP group (OR = 2.152, 95% CI: 1.566–2.957), had fewer male sexual partners in the last two weeks (OR = 0.685, 95% CI: 0.563–0.834), were one year older on average (OR = 1.022, 95% CI: 1.002–1.043), considered that medication kept them safe, were less worried about others knowing they took medicine and were more likely to have high adherence. *Conclusions*: The efficacies of daily TDF and event-driven TDF use were not significantly different in preventing new infections among HIV-negative MSM. Event-driven TDF use is economical and effective and is worth popularizing. Our results provide evidence for the application and promotion of the PrEP strategy in the MSM population in China.

## 1. Introduction

Acquired immunodeficiency syndrome (AIDS) is an important global public health problem. By the end of 2017, 36.9 million people were living with human immunodeficiency virus (HIV) infection worldwide, with 1.8 million new infections and 940,000 AIDS-related deaths each year [1]. In China, among the newly reported HIV/AIDS cases, the proportion of MSM has been increasing yearly. The proportion of HIV infections transmitted through MSM increased from 3.4% in 2007 to 28% in 2017, and currently, HIV infection is spreading extremely rapidly among MSM in China [2]. Thus, the HIV infection situation is serious in this population and needs effective intervention and control.

Behavioral intervention such as condom use has been proved effective in preventing new HIV infections, however, the effectiveness of single behavioral intervention is not obvious, and needs the combined biomedical interventions such as PrEP. Currently, pre-exposure prophylaxis (PrEP) is an important biological prevention strategy among MSM. The PrEP regimen mainly includes the use of tenofovir disoproxil fumarate (TDF) monotherapy or a combination of TDF and emtricitabine (FTC) (also known as Truvada) [3]. The effectiveness and safety of a long-term use of TDF or TDF/FTC for preventing new HIV-1 infections have both been confirmed in healthy people [4,5,6,7,8]. However, no comparative studies have been conducted on the use of TDF monotherapy in the MSM population; prior studies have focused on comparing between daily TDF and TDF/FTC regimens or between event-driven TDF/FTC and daily TDF/FTC regimens among MSM. Since 2018, the China Center for Disease Control and Prevention (CDC) has attempted to popularize TDF/FTC in seven provinces. PrEP has not been popularized in China, and a low level of PrEP awareness has been reported in the Chinese MSM population [9]. Hence, it is valuable to explore the effectiveness and subjects’ adherence pertaining to two intervention methods in preventing new HIV infections in the daily PrEP group and event-driven group.

This study aimed to investigate the factors influencing medication adherence, with a focus on the effect of a medication regimen upon adherence. The results from the study will inform the choice of medication regimen and strategies to promote PrEP among MSM in China.

## 2. Subjects and Methods

### 2.1. Research Subjects

HIV-negative men who have sex with men (MSM) people were recruited into the study by convenience sampling at four research centers in Chongqing, Guangxi, Xinjiang and Sichuan. Recruitment was performed using the following methods: first, through non-governmental organizations (NGOs) which were established voluntarily by the MSM population, they were mainly responsible for recruitment and cohort maintenance; second, through peer introduction, and by the dissemination of information through announcements at public places and at places of entertainment frequented by the MSM population (usually park, public bathhouse, bar); third, through voluntary counseling and testing for AIDS (VCT) clinics; fourth, through the Internet.

Inclusion criteria were as follows: (1) signed informed consent; (2) age ≥ 18 and ≤ 65 years (3) HIV antibody-negative; (4) participated in sexual intercourse at least once every two weeks; (5) at least one homosexual partner within a month before the trial; (6) willing to use the study medication under guidance and obey follow-up arrangements; (7) willing to participate in the trial for 96 weeks.

### 2.2. Trial Contents and Methods

#### 2.2.1. Design

The study was an open, non-randomized, multicenter, parallel controlled clinical intervention study based on standard AIDS prevention interventions (study time: April 2013 to March 2015, registration number: ChiCTR-TRC-13003849). The subjects at each research center were assigned a random number and divided in a 1:1:1 ratio into each of the three groups: daily PrEP group, event-driven group and blank control group. A random number was generated by Statistical Analysis System (SAS) 9.4 software program (SAS Institute, Cary, NC, USA), whereupon each third of these subjects were respectively divided into the three groups using an ascending random number method. The daily PrEP group was given oral TDF (300 mg daily) (produced and provided by Gilead Sciences, Inc. (Foster City, CA, USA), specifications: 300 mg/per tablet. Lot: A818213). Subjects in the event-driven group took 300 mg TDF orally 48–24 h before sexual activity, and 300 mg TDF 2 h after sexual activity. The dosage was no more than 300 mg within 24 h. The blank control group was being organized, and did not receive any drugs or placebos. All subjects underwent HIV testing and counseling on reducing the risk of HIV infections, and received free condoms and standard AIDS prevention intervention services on the management of sexually transmitted diseases. However, subjects in Chongqing and Sichuan were completely randomized, whereas those in Xinjiang and Guangxi were allowed to voluntarily choose between daily or event-driven PrEP regimens in the practical operation, because the clinical trial was conducted by the local CDC and not the research center, therefore some subjects did not honor the grouping arrangement and entered their respective groups by self-selection. Subjects in the free-choice sites and random allocation sites were pooled for larger sample size in analyses.

#### 2.2.2. Study Procedures

(1) Screening: An optimal screening process was established at each research center; screening numbers were complied, and subjects were selected according to established inclusion and exclusion criteria. (2) Cohort study initiation: Enrolled subjects were included within eight weeks; baseline clinical and laboratory tests such as HIV-1 serological tests, hepatitis B virus (HBV) serological tests, and blood biochemical tests were conducted after obtaining subjects’ informed consent. (3) Follow-up during the medication period: face-to-face follow-up of subjects in two groups was conducted every 12 weeks to collect data on high-risk behavior and medication in the recent two weeks. Clinical and laboratory tests including an HIV test, blood biochemical examination, urine test and so on were also conducted. Subjects’ serum was obtained and stored; the sample was collected before using medication on the day of follow-up. (4) Evaluation of serum TDF concentration: serum TDF concentrations in men in Chongqing and Sichuan were quantitated from stored subject samples. For subjects with conversion to HIV-positive status, all serum samples obtained from enrolment to detection of HIV-positive status were processed; among those who remained HIV-negative, subjects with at least one serum sample collected during the study period were selected by systematic sampling (after excluding subjects who were HIV-positive and who refused to undergo follow-up, subjects were selected every other line in the database when sorted by CRF number), all serum samples of selected HIV-negative subjects were tested.

#### 2.2.3. Measures

The outcome measures mainly included the effectiveness of TDF regimens and the adherence of MSM. The effectiveness evaluation index was obtained based on the HIV incidence; the adherence difference evaluation index was evaluated based on the average serum concentration of TDF; the outcome variable index of adherence-influencing factors was evaluated using the medication adherence rate, which was calculated as follows: medication adherence rate = 100% − (missed medication/medication that should have been taken) × 100%; medication that the event-driven group should have taken = number of episodes of sexual intercourse × 2, and the missed medication was assessed by self-report. 

The influencing factors for adherence were evaluated using cross-sectional baseline data and data obtained at the first follow-up for each subject. The adherence level in the first follow-up was used as a dependent variable. A medication adherence rate <80% indicated low adherence and a medication adherence rate >80% indicated high adherence. Behavioral characteristics included HIV counseling history, HIV testing history, frequency of sexual partners through the Internet, sexually transmitted disease (STD) history, drinking frequency, history of drug use and history of using commercial sexual services; psychological characteristics and attitudes of taking medicine included side effects, risk perception and efficacy cognition. The research methods primarily included cohort follow-up and laboratory testing. Cohort follow-up was conducted using self-completed questionnaires which were distributed to subjects. Data on individual demographic information, medication psychology, and self-reported sexual behavior in the last half month were collected at baseline and after every 12 weeks during follow-up; laboratory tests included routine tests, virological tests and serum TDF concentration monitoring. Sample collection methods were as follows: within one hour after 5 mL of venous blood was obtained from the subjects by vacuum blood collection, the blood samples were centrifuged in a test tube to separate the serum from the blood cells; 1mL of serum from each sample was immediately transferred to two cryopreserved tubes and stored at −70 °C or lower. Samples with hemolysis and those containing blood cells could not be analyzed. TDF concentration was quantitated by high performance liquid chromatography with ultraviolet detection (HPLC-UV) after solid phase extraction. In addition to the above data, individual safety was assessed and documented, which including information on the timing, severity, duration, measures taken and outcome of adverse events.

### 2.3. Quality Control and Ethics

The entire trial process was in conformity with the guidelines from the Helsinki Declaration and the Chinese clinical trial research norms and regulations, and was approved by the Ethics Committee of Chongqing Medical University (Ethical Approval code: 2012010). Subjects were informed of clinical review, and the privacy and data of the subjects was strictly protected. The inspectors and researchers involved in the study were trained to thoroughly inspect the work of the clinical research center regularly and to specify the verification results of the original data in accordance with standard operating procedures. Verification of informed consent for all subjects was mandatory.

### 2.4. Statistical Analysis

EpiData 3.0 software (EpiData Associations, Odense, Denmark) was used for data double entry and verification and SAS9.4 was used for statistical analysis. Descriptive statistics was used to analyze the basic data of the daily PrEP group and the event-driven group; the log-rank test was used to compare differences in overall HIV incidence between two groups; the relative risk (RR) value was calculated by the Woolf method, the daily PrEP group was considered the reference group; the rank test was used to compare the mean blood drug concentration of the two groups. Among univariate analysis, a χ^2^ test was conducted for analyzing the medication adherence, the variables that included behavioral characteristics, the psychological characteristics of taking medicine and AIDS-related attitudes. The logistic, stepwise regression model was used in multivariate analysis, the criterion for variable entry and removal was 0.05, and the odds ratio (OR) value and the 95% confidence interval (CI) were calculated. The medication adherence rate used in the HIV incidence and serum TDF concentration analysis was computed with full observation period data, while univariate and multivariate analysis was computed using first follow-up visit data. Furthermore, missing data were removed from statistical analyses, and *p* < 0.05 indicates statistical significance.

## 3. Results

### 3.1. Subject Characteristics

A total of 2422 subjects were screened for eligibility, and 1884 subjects were randomized. After randomization, there were 408 subjects each in the daily PrEP and event-driven groups in the random allocation sites; in the free-choice sites, there were 167 subjects in the daily PrEP group and 253 in the event-driven group; 648 subjects in the blank control group were not included in our analysis. Among those excluded before randomization, 376 were HIV-positive, 72 were Hepatitis C positive, 60 declined to participate, 30 were age <18 or >65 years old. Among those excluded after randomization, in the random allocation sites, 60 were asexual or did not provide information of sexual behavior within 96 weeks, and 241 refused to undergo follow-up examinations; in the free-choice sites, 49 were asexual or did not provide information, and 30 refused to undergo follow-up examinations. A total of 856 valid subjects were selected after screening. Figure 1 presents the subject flowchart per the CONSORT guidelines. In addition, after excluding subjects who were HIV-positive (*N* = 39) and who also refused to undergo follow-up examinations (*N* = 241) in the random allocation sites, 268 of the 536 HIV-negative subjects were selected by systematic sampling to undergo serum TDF concentration testing. During the study, no serious adverse events or unintended effects occurred in either group.

The overall average age of the subjects was 30.44 years (median age 29 years); the Han nationality accounted for 91.4% of the study subjects; 74.4% held urban household registrations; 39.5% were educated to the university level or above; 11.2% were educated to the junior high school and below level; the majority of the subjects were employed, accounting for 79.0% of the study population; the average monthly income was below 1000 yuan (ca 142 USD) for 14.6% of the subjects; it was 1000–3000 yuan for 35.9% of the subjects, and was 3000–5000 yuan for 36.9% of the subjects. The abovementioned basic demographic and behavioral characteristics were not statistically different between the two groups (*p* > 0.05), similar to that of the two groups in the random allocation and free-choice sites (Table 1). Meanwhile, the number of drugs that ought to have been taken was 18,606 in the daily PrEP group and 8166 in the event-driven group during the full observation period. Average follow-up times is 3.865, 4.093 separately in the daily PrEP and event-driven PrEP group. Among the baseline subject characteristics, age <30 years, being of Han nationality, and not having access to free HIV consultation and testing, were significantly more frequent in the 380 subjects who were excluded than in the 856 valid subjects.

### 3.2. Sero-Conversion in Subjects

Positive sero-conversion was observed in 30 of 385 subjects in the daily PrEP group and in 32 of 471 subjects in the event-driven group. The overall HIV incidence was 6.60 cases/100 person-years in the daily PrEP group and 5.565 cases/100 person-years in the event-driven group. The overall RR with 95% CI was 0.844 (0.492–1.449). On performing the log-rank test using the life-table method, the log-rank statistic was found to be 0.168 (*p* > 0.05). In addition, Table 2 shows the HIV incidence stratified by medication adherence and research centers, and when stratified by medication adherence, 22 subjects were excluded from analysis for lacking a medication adherence rate; among those excluded, 18 were HIV-negative (4 in the event-driven, and 14 in the daily PrEP groups) and 4 were HIV-positive in the daily PrEP group. HIV incidence did not differ significantly between the two groups when stratified by medication adherence and site (Table 2 and Figure 2). Meanwhile, the overall HIV incidence in the blank control group was 6.175 cases/100 person-years. The overall HIV incidence did not differ significantly between the three groups (*p* > 0.05), however, when medication adherence was ≥80%, HIV incidences were both significantly lower in the two groups than in the blank control group (*p* < 0.05).

### 3.3. Serum TDF Concentration

We quantified 268 serum samples from HIV-negative subjects and 39 serum samples from HIV-positive subjects in random allocation sites (151 and 156 subjects in the daily PrEP and event-driven groups, respectively), where 46 subjects were excluded in the analysis (9 lacked their medication adherence rate and 37 were asexual or did not provide information of sexual behavior), and we finally evaluated 261 serum samples for TDF concentrations (124 and 137 subjects in the daily PrEP and event-driven groups, respectively). When the medication adherence rate was <80%, the overall median (P25–P75) serum concentrations were 0.404 (0.237–0.661) mg/L and 0.472 (0.255–0.987) mg/L in the daily PrEP and event-driven groups, respectively. When the medication adherence rate was ≥80%, the overall median (P25–P75) serum concentrations were 0.458 (0.272–0.625) mg/L and 0.429 (0.278–0.648) mg/L in the daily PrEP and event-driven groups, respectively. Serum TDF concentrations did not differ significantly between the two groups on stratification by medication adherence (*p* > 0.05) (Table 3).

### 3.4. Factors Influencing Adherence

#### 3.4.1. Univariate Analysis

In terms of behavioral characteristics, the overwhelming majority of the subjects had been tested for HIV, and 17.2% (137/795) had not been tested for HIV; 65.2% (515/790) had received HIV counseling before the intervention; 7.1% (54/756) had frequently searched for sexual partners through the Internet; the overwhelming majority of the population had not been diagnosed with HIV in the last six months, which accounted for 91.2% (721/791) of the study population; 94.8% (750/791) of the individuals did not utilize commercial sexual services in the last six months; 96.9% (755/779) of the individuals did not use drugs in the last six months. In the category of perceptions of taking drugs and AIDS-related attitudes, the items on the questionnaires and the results obtained were as follows: “Drug makes me safer, and keep me away from AIDS”, 27.6% (218/789) subjects disagreed; “I was worried about others knowing I was taking medication”, 30.1% (238/790), 26.8% (212/790), 18.8% (148/790), 7.1% (56/790) and 17.2% (136/790) selected “completely not worried” to “very worried” respectively; “I was worried about the side effects of the medicine”, 19.0% (150/789), 27.1% (214/789), 25.0% (197/789), 10.4% (82/789) and 18.5% (146/789) selected “completely not worried” to “very worried”, respectively. 

Results of the χ^2^ tests revealed that men who considered that the medication kept them safe, had a doctor-diagnosed STD history, were less worried about side effects or the ineffectiveness of the medicine, along with awareness by others of their use of this medication, and were more likely to have higher adherence than other men (*p* < 0.05) (Table 4).

#### 3.4.2. Multivariate Logistic Regression Analysis

Medication adherence rates computed with first follow-up visit data were taken as a dependent variable, while the independent variables were as follows: medication regimen; demographic characteristics including age, household registration, education, employment, marital status and monthly income; behavioral characteristics; psychological characteristics of taking medication; sexual behavior characteristics including sexual partners, number of episodes of sexual behavior and frequency of condom use. The variables with *p* < 0.15 in Table 4, as well as the medication regimen, demographic characteristics and sexual behavior characteristics were included as independent variables in the logistic stepwise regression model. The results showed that subjects who were in the event-driven PrEP group (OR = 2.152, 95% CI: 1.566–2.957), had fewer male sexual partners in the last two weeks (OR = 0.685, 95% CI: 0.563–0.834), were one year older on average (OR = 1.022, 95% CI: 1.002–1.043), considered that medication kept them safe, were less worried about others knowing they took medicine and were more likely to have high adherence. Data from subjects who disagreed with the above were taken as reference variables, and in the “medicine keeps me safer, and keeps me away from AIDS” item, each level compared with those subjects who disagreed had statistical significance. Adherence was better in people who agreed with the above item than in those who completely disagreed with this view; on the issue of “I am worried that others know I am taking medication”, adherence was lower in people who were relatively or very worried than it was in those who were completely not worried. Meanwhile, according to the result of sub-analysis, in the random allocation sites, subjects who were one year older on average (OR = 1.044, 95% CI: 1.018–1.070), had fewer male sexual partners in the last two weeks (OR = 0.660, 95% CI: 0.522–0.835), and were more likely to have high adherence. In the free-choice sites, subjects who were in the event-driven PrEP group (OR = 13.137, 95% CI: 6.664–26.614) considered that medication kept them safe, were little more worried about others knowing they took medicine and were more likely to have high adherence (Table 5).

## 4. Discussion

As early as 2012, the United States (U.S.) Food and Drug Administration (FDA) approved daily TDF/FTC combined therapy for the prevention of new HIV infections [10]. Although the World Health Organization (WHO) and the US Centers for Disease Control and Prevention (CDC) still recommend daily PrEP medication for people at high risk of AIDS, the latest guidelines released by both the French and the European Clinical Societies for AIDS recommend TDF/FTC combined therapy before and after sexual behavior [11]. However, there are few studies on the use of TDF monotherapy in HIV prevention by on-demand and daily regimens. This study is based on the PrEP medication regimen of the MSM population in Western China, and primarily evaluated the effectiveness of and adherence to medication regimens. The results showed that there was no statistical difference in TDF serum concentration between the two groups after stratification according to adherence, and HIV incidence was not statistically different between the groups during the 96-week clinical intervention. The factors influencing the adherence of this group included medication regimen, age, the number of sexual partners in the last two weeks, the individual’s cognition of drug effectiveness and worry that others knew that they were taking medication.

In terms of effectiveness, the overall HIV incidence was 6.60 cases/100 person-years in the daily PrEP group, while it was 5.57 cases/100 person-years in the event-driven group. When adherence was high, the HIV incidence in the daily PrEP group dropped to 2.72 cases/100 person-years and to 1.817 cases/100 person-years in the event-driven group. There was no statistical difference in HIV incidence between the two groups when stratified by sites or adherence. Moreover, according to serum samples collected from centers in Chongqing and Sichuan, there was no statistically significant difference in the mean TDF serum concentration between the two groups. 

It can thus be concluded that on-demand taking of TDF has the same effect as taking TDF orally and daily to prevent new HIV infections in the Chinese MSM population. Additionally, for the reason that subjects in the event-driven PrEP group take medicine based on their sexual activity, the number of tablets using the event-based regimen was potentially smaller compared to that using the daily TDF regimen actually. This aspect may be used to reduce the economic burden of those taking this drug and improve cost-effectiveness [12]. Although the Ipergay study in France researched and confirmed the effectiveness of on-demand PrEP [13], it also indicated that the efficacy of on-demand PrEP to prevent new HIV infections depends upon the accuracy of an individual’s prediction of their future sexual behavior. When the accuracy of the individual’s prediction was low, the efficacy of on-demand PrEP prevention was also limited [14], which may be one of the reasons for the higher overall HIV incidence in the event-driven group compared to that in the daily PrEP group.

In accordance with results obtained from research conducted worldwide, the level of adherence between two groups determines the effectiveness of prevention, and subjects in high adherence groups have a lower attack rate [15]. Therefore, strengthening the drug adherence of the MSM population and achieving precise interventions is important and worthy of attention. In this study, cross-sectional data from the first follow-up of each subject were used to for analysis in the conjunction adherence level as a dependent variable. The results showed that the adherence of the subjects in the event-driven group was better than that in the daily PrEP group (OR = 2.152). If the daily TDF group did not have daily sexual behavior, it is possible that subjects in that group may have assumed that there is no need to take the medicine without daily sexual behavior, which may potentially have increased the missing rate. Subjects in the event-driven group only needed to take the medicine before and after sexual activity, and the likelihood of missing doses was thus relatively low in this group. The TDF serum concentration of the daily TDF group was theoretically higher than that of the event-driven group which had a medication regimen based on sexual behavior. However, the results of this study showed no statistically significant differences in the serum TDF concentration between the two groups, which indirectly proved that the adherence of the event-driven group was higher. In subgroup analysis of subjects in random sites, MSM participants in the event-driven group showed a higher adherence but the difference was not statistically significant, and this could be caused by the small sample size; furthermore, despite the lack of differences in adherence between the two regimens, we may still recommend event-driven PrEP owing to similar effectiveness and fewer costs compared with daily PrEP. Subjects who were one year older on average had higher adherence (OR = 1.022), which was consistent with that reported by several existing international studies. This was especially true for antiretroviral therapy, and younger HIV-infected individuals showed lower adherence than did older HIV-infected individuals. The possible reason for this observation may be that older people have a more stable life and a higher positive expectation of the drugs. More attention ought to be paid to Younger MSM in a further large-scale intervention study [16,17,18]. A greater the likelihood of high adherence was associated with a lower number of male sexual partners in the past two weeks (OR = 0.685). This may be because when there were fewer partners, the individual self-safety cognition and risk perception level was high. Conversely, with an increase in the number of sexual partners, indicating a greater randomness in life, adherence may have decreased. Studies have shown that adherence is generally low when the number of sexual partners exceeds four, so the intervention should also be strengthened in multi-partner populations; subjects with more than four sexual partners should be included as the focus of the intervention [19]. When individuals are more worried about others knowing that they are taking medicines and additionally think that the medicines do not work, the likelihood of missing doses increases. This is also in line with the theory of knowledge, belief and practice. Unfavorable opinion of PrEP and fearing the stigma caused by the discovery of medicine use by others can cause low adherence in order to avoid social discrimination. For this reason, researchers should work towards increasing the effectiveness of these medicines. Additionally, disseminating proper information about the effectiveness of these regimens by peer education among MSM subjects will help to reduce their psychological burden, increase self-identity and improve adherence in follow-up trials [20]. 

Lastly, this clinical trial was conducted in western China, because the MSM population in western China was concentrated and had high HIV prevalence in the past years, therefore they could be a representative MSM population in China.

## 5. Limitations

This study had several limitations. First, this study was not exactly a randomized trial, though it was designed as a completely randomized clinical intervention study. It could be called as a real-world study, which includes sites with randomization and sites where the subjects are not randomized. However, for a large sample size, we included data from the free-choice sites in the analysis. Second, the lack of significant differences in HIV incidence rates between the two groups did not mean that the two groups had the same incidence rates; however, as 1.612 cases/100 person-years was the maximum difference in HIV incidence rates, we may conclude that the two regimens show similar effectiveness for preventing new HIV infections. Third, the number of self-reported missing doses and serum concentration test results were used to evaluate subjects’ adherence to the regimen. The advantages of self-reporting are that obtaining information is easy and convenient; however, the disadvantages of this method include susceptibility to subjective influences, such as intentional underreporting or recall bias. The advantage of serum concentration testing is that it objectively reflects drug intake in the subjects, but these drug concentrations are influenced by factors such as drug half-life; meanwhile, we compared serum TDF concentration stratified by medication adherence in the two groups, and both showed no significant difference (*p* > 0.05). Thus, serum levels may not accurately reflect long-term drug intake in the subjects. In subsequent clinical trials, for better accuracy, drug concentrations can be measured in hair samples or peripheral blood cells or electronic bottle cap levels can be used as adherence indicators [21].

## 6. Conclusions

Our study on the implementation of PrEP therapy in a Chinese MSM population showed that, compared to daily TDF use, a TDF-only regimen (on-demand PrEP) was associated with good effectiveness and adherence. For promoting PrEP in the future, event-driven PrEP can be an economic and effective option.

## Figures and Tables

**Figure 1 ijerph-16-04996-f001:**
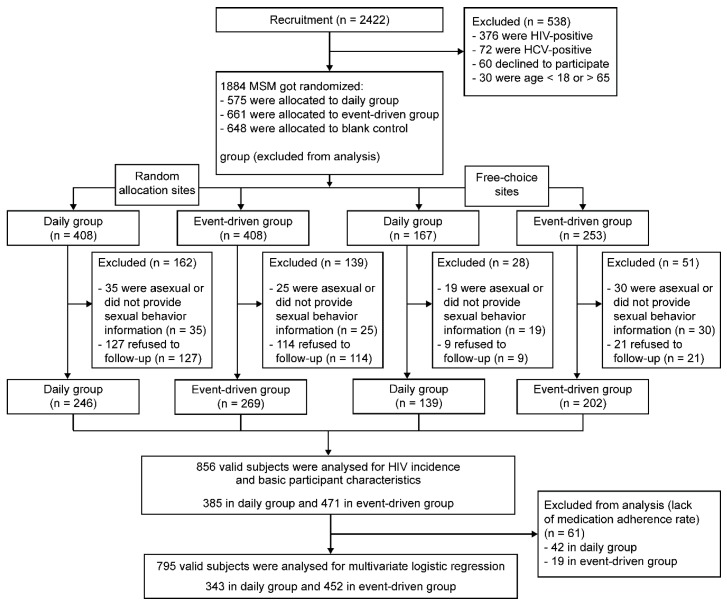
Flow diagram for the clinical trial.

**Figure 2 ijerph-16-04996-f002:**
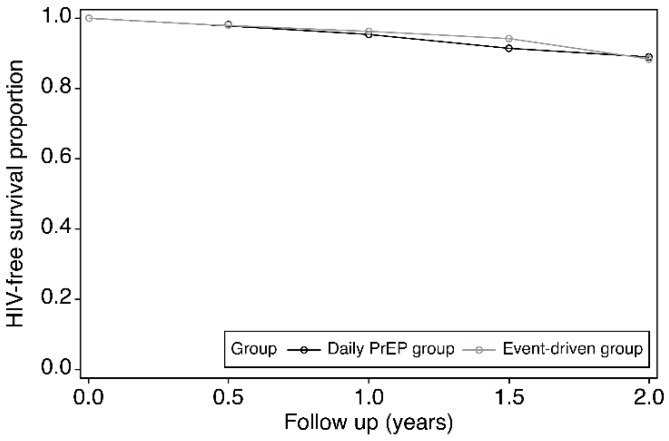
Life-table survival curve of overall subjects.

**Table 1 ijerph-16-04996-t001:** The socio-demographic characteristics and behaviors of men who have sex with men (MSM).

Variable	Daily PrEP Group (*N* = 385)	Event-Driven Group (*N* = 471)	χ^2^ Value	*p* Value	Daily PrEP Group in Random Allocation Sites (*N* = 246)	Event-Driven Group in Random Allocation Sites (*N* = 269)	χ^2^ Value	*p* Value	Daily PrEP Group in Free-Choice Sites (*N* = 139)	Event-Driven Group in Free-Choice Sites (*N* = 202)	χ^2^ Value	*p* Value
Age (years)			2.08	0.149			1.01	0.314			1.04	0.308
<30	139 (36.1%)	31.4 (51.2%)			89 (36.2%)	86 (31.97%)			50 (36.0%)	62 (30.7%)		
Household registration *			1.09	0.296			0.12	0.730			0.60	0.439
Town	279 (72.7%)	357 (75.8%)			165 (67.4%)	185 (68.8%)			114 (82.0%)	172 (85.2%)		
Nationality			0.18	0.674			0.31	0.580			0.16	0.690
Han nationality	353 (91.7%)	428 (90.9%)			239 (97.2%)	259 (96.3%)			114 (82.0%)	169 (83.7%)		
Degree of Education *			1.91	0.592			3.64	0.303			1.10	0.778
Junior high school and below	41 (10.7%)	55 (11.7%)			31 (12.6%)	42 (15.6%)			10 (7.3%)	13 (6.4%)		
Senior high school/vocational high school/technical secondary school	109 (28.4%)	114 (24.3%)			80 (32.5%)	70 (26.0%)			29 (21.0%)	44 (21.9%)		
Junior College	87 (22.7%)	111 (23.6%)			58 (23.6%)	60 (22.3%)			29 (21.0%)	51 (25.4%)		
University or above	147 (38.3%)	190 (40.4%)			77 (31.3%)	97 (36.1%)			70 (50.7%)	93 (46.3%)		
Employment *			0.10	0.949			3.12	0.211			0.05	0.978
Employed	293 (76.7%)	281 (80.9%)			185 (75.8%)	221 (82.1%)			108 (78.3%)	160 (79.2%)		
Students	54 (14.1%)	54 (11.4%)			36 (14.8%)	29 (10.8%)			18 (13.0%)	25 (12.4%)		
Unemployed or retired	35 (9.2%)	36 (9.7%)			23 (9.4%)	19 (7.1%)			12 (8.7%)	17 (8.4%)		
Marital status			5.09	0.079			2.11	0.348			5.13	0.077
Unmarried	297 (77.1%)	336 (71.4%)			182 (74.0%)	184 (68.4%)			115 (82.7%)	152 (75.2%)		
Married	61 (15.9%)	83 (17.6%)			42 (17.1%)	53 (19.7%)			19 (13.7%)	30 (14.9%)		
Divorced or widowed	27 (7.0%)	52 (11.0%)			22 (8.9%)	32 (11.9%)			5 (3.6%)	20 (9.9%)		
Monthly income *			1.99	0.738			7.57	0.109			6.66	0.155
1000 yuan or below	57 (15.1%)	66 (14.2%)			43 (17.7%)	34 (12.7%)			14 (10.3%)	32 (16.1%)		
1001~3000 yuan	138 (36.4%)	165 (35.4%)			98 (40.3%)	113 (42.3%)			40 (29.4%)	52 (26.1%)		
3001~5000 yuan	132 (34.8%)	180 (38.6%)			76 (31.3%)	92 (34.5%)			56 (41.2%)	88 (44.2%)		
5001~10,000 yuan	44 (11.6%)	44 (9.4%)			24 (9.9%)	19 (7.1%)			20 (14.7%)	25 (12.6%)		
above 10,000 yuan	8 (2.1%)	11 (2.4%)			2 (0.8%)	9 (3.4%)			6 (4.4%)	2 (1.0%)		
Have you taken the initiative to conduct HIV free consultation? *			0.46	0.496			0.26	0.608			0.37	0.541
Yes	243 (63.3%)	306 (65.5%)			128 (52.2%)	145 (54.5%)			115 (82.7%)	161 (80.1%)		
Have you tested the HIV virus?			0.01	0.959			0.04	0.848			2.20	0.138
Yes	70 (18.2%)	85 (18.1%)			64 (26.0%)	68 (25.3%)			6 (4.3%)	17 (8.4%)		
Frequency of sexual partners through the Internet *			2.50	0.476			0.54	0.911			2.93	0.402
Often	23 (6.3%)	34 (7.5%)			16 (7.1%)	20 (7.9%)			7 (5.1%)	14 (7.0%)		
Sometimes	66 (18.1%)	75 (16.6%)			39 (17.2%)	39 (15.5%)			27 (19.7%)	36 (17.9%)		
Occasional	137 (37.7%)	190 (41.9%)			87 (38.3%)	102 (40.5%)			50 (36.5%)	88 (43.8%)		
Completely none	138 (37.9%)	154 (34.0%)			85 (37.4%)	91 (36.1%)			53 (38.7%)	63 (31.3%)		
Have you been diagnosed with sexually transmitted diseases by doctors? *			1.71	0.191			2.95	0.086			0.16	0.685
Yes	38 (10.0%)	35 (7.5%)			21 (8.6%)	13 (4.9%)			17 (12.3%)	22 (10.9%)		
Drinking alcohol frequency in a recent month *			3.69	0.449			4.91	0.297			2.73	0.604
Daily	13 (3.4%)	21 (4.5%)			11 (4.6%)	16 (6.0%)			2 (1.5%)	5 (2.5%)		
At least three times/week	39 (10.2%)	42 (8.9%)			30 (12.2%)	23 (8.6%)			9 (6.5%)	19 (9.4%)		
At least once/week	55 (14.4%)	81 (17.2%)			30 (12.2%)	36 (13.4%)			25 (18.1%)	45 (22.3%)		
Less than once a week	146 (38.1%)	189 (40.1%)			86 (35.1%)	112 (41.6%)			60 (43.5%)	77 (38.1%)		
No alcohol	130 (33.9%)	138 (29.3%)			88 (35.9%)	82 (30.4%)			42 (30.4%)	56 (27.7%)		
Drug use in the last half a year *			0.83	0.362			0.99	0.319			0.04	0.852
Using	9 (2.4%)	16 (3.5%)			4 (1.7)	8 (3.0)			5 (3.7%)	8 (4.1%)		
Commercial sex behavior in recent half a year *			0.14	0.710			0.02	0.883			0.59	0.441
Occurring	20 (5.2%)	22 (4.7%)			12 (4.9%)	14 (5.2%)			8 (5.8%)	8 (4.0%)		

Note: * indicates missing data.

**Table 2 ijerph-16-04996-t002:** HIV incidence for MSM by adherence and sites subgroups.

Stratification Factors		Group	Positive Conversion	The Number of Exposed Subjects	Total Person Years at Risk	Overall Person Years	HIV Incidence (Cases/100 Person-Years)	RR Vale (95% CI)
Overall		Daily PrEP group	30	385	26.75	454.5	6.60	0.844 (0.492–1.449)
		Event-driven group	32	471	29.00	575.0	5.565	
Medication adherence rate	<80%	Daily PrEP group *	22	243	20.00	294.75	7.464	1.216 (0.661–2.236)
		Event-driven group *	27	252	25.50	297.5	9.076	
	≥80%	Daily PrEP group *	4	124	3.00	147.0	2.720	0.668 (0.141–3.164)
		Event-driven group *	5	215	3.50	275.25	1.817	
Sites	Random allocation sites	Daily PrEP group	20	246	20.50	297.0	6.734	0.844 (0.425–1.675)
		Event-driven group	19	269	15.00	334.25	5.684	
	Free-choice sites	Daily PrEP group	10	139	6.25	157.50	6.349	0.851 (0.324–2.233)
		Event-driven group	13	202	14.00	240.75	5.400	

Note: * indicates missing data.

**Table 3 ijerph-16-04996-t003:** Serum tenofovir disoproxil fumarate (TDF) concentration of MSM in Chongqing and Sichuan.

Medication Adherence Rate	Medication Pattern	N	Median(mg/L)	P25–P75(mg/L)	Z	*p*
<80%	Daily PrEP group *	67	0.404	0.237–0.661	−0.997	0.319
	Event-driven group *	71	0.472	0.255–0.987		
≥80%	Daily PrEP group *	57	0.458	0.272–0.625	0.190	0.849
	Event-driven group *	66	0.429	0.278–0.648		

Note: * indicates missing data.

**Table 4 ijerph-16-04996-t004:** Univariate analysis of the adherence among MSM.

Variable	High Adherence (*N* = 409)	Low Adherence (*N* = 386)	χ^2^	*p*
Behavioral characteristics of subjects				
Have you taken the initiative to conduct HIV free consultation? *			2.22	0.136
Yes	274 (67.7%)	241 (62.6%)		
Have you tested the HIV virus?			0.01	0.928
Yes	339 (82.9%)	319 (82.6%)		
Frequency of sexual partners through Internet in recent half a year *			5.49	0.139
Often	31 (7.9%)	23 (6.3%)		
Sometimes	66 (16.8%)	58 (15.9%)		
Occasional	169 (43.1%)	137 (37.7%)		
Completely none	126 (32.2%)	146 (40.1%)		
Have you been diagnosed with sexually transmitted diseases by doctors in recent half a year? *			2.73	0.099
Yes	29 (7.1%)	40 (10.4%)		
Drinking alcohol frequency in a recent month *			6.44	0.169
Daily	18 (4.4%)	15 (3.9%)		
At least three times/week	41 (10.0%)	33 (8.6%)		
At least once/week	56 (13.7%)	70 (18.2%)		
Less than once a week	155 (37.9%)	160 (41.7%)		
No alcohol	139 (34.0%)	106 (27.6%)		
Drug use in the last half a year *			0.03	0.873
Using	12 (3.0%)	12 (3.2%)		
Commercial sex behavior in recent half a year *			0.35	0.552
Occurring	23 (5.6%)	18 (4.7%)		
Perceptions and AIDS related attitudes of the subjects				
What do you think of the severity of AIDS? *			4.89	0.299
Very high	266 (65.0%)	278 (72.2%)		
Relatively high	118 (28.9%)	88 (22.9%)		
Average	20 (4.9%)	16 (4.2%)		
Relatively low	3 (0.7%)	2 (0.5%)		
Very low	2 (0.5%)	1 (0.2%)		
Medicine makes me safe, away from AIDS *			19.93	0.001
Completely none	89 (21.9%)	129 (33.7%)		
A little	86 (21.2%)	88 (23.0%)		
Somewhat	98 (24.2%)	61 (15.9%)		
Majority	46 (11.3%)	44 (11.5%)		
Always	87 (21.4%)	61 (15.9%)		
I am worried that the medicine does not work *			13.03	0.011
Completely none	96 (23.7%)	89 (23.2%)		
A little	148 (36.5%)	120 (31.3%)		
Somewhat	87 (21.4%)	64 (16.7%)		
Majority	20 (4.9%)	34 (8.9%)		
Always	55 (13.5%)	76 (19.9%)		
I’m worried about the side effects of medicine *			17.74	0.001
Completely none	92 (22.6%)	58 (15.1%)		
A little	121 (29.8%)	93 (24.3%)		
Somewhat	99 (24.4%)	98 (25.6%)		
Majority	34 (8.4%)	48 (12.5%)		
Always	60 (14.8%)	86 (22.5%)		
I am worried that others know that I am taking medicine *			15.58	0.004
Completely none	136 (33.4%)	102 (26.6%)		
A little	120 (29.5%)	92 (24.0%)		
Somewhat	74 (18.2%)	74 (19.3%)		
Majority	21 (5.2%)	35 (9.2%)		
Always	56 (13.7%)	80 (20.9%)		
I felt the side effects of medicine *			5.97	0.201
Completely none	238 (58.6%)	241 (62.9%)		
A little	115 (28.3%)	92 (24.0%)		
Somewhat	38 (9.4%)	27 (7.1%)		
Majority	9 (2.2%)	11 (2.9%)		
Always	6 (1.5%)	12 (3.1%)		
I am more afraid of AIDS *			12.05	0.017
Completely none	218 (53.6%)	200 (52.2%)		
A little	117 (28.8%)	113 (29.5%)		
Somewhat	40 (9.8%)	35 (9.2%)		
Majority	4 (1.0%)	18 (4.7%)		
Always	28 (6.8%)	17 (4.4%)		
I feel that the doctors have discriminated against me *			0.35	0.987
Completely none	372 (91.4%)	346 (90.3%)		
A little	20 (4.9%)	21 (5.5%)		
Somewhat	6 (1.5%)	6 (1.6%)		
Majority	4 (1.0%)	5 (1.3%)		
Always	5 (1.2%)	5 (1.3%)		

Note: * indicates missing data.

**Table 5 ijerph-16-04996-t005:** Multivariate logistic stepwise regression of adherence among MSM.

Variable (Controlled Group)	β	*p* Value	OR (95% CI)
Overall analysis			
Age *	0.0221	0.0290	1.022 (1.002–1.043)
Number of sexual partners in the last two weeks *	−0.3783	0.0002	0.685 (0.563–0.834)
Medication regimen (daily PrEP group) *			
event-driven group	0.7662	<0.0001	2.152 (1.566–2.957)
Medicine makes me safe, away from AIDS (Completely none) *			
A little	0.5462	0.0163	1.727 (1.106–2.696)
Somewhat	1.1155	<0.0001	3.051 (1.921–4.847)
Majority	0.5973	0.0325	1.817 (1.051–3.141)
Always	0.8935	0.0002	2.444 (1.522–3.924)
I am worried that others know I am taking medicine *			
A little	0.1412	0.5064	1.152 (0.759–1.746)
Somewhat	−0.0651	0.7780	0.937 (0.596–1.473)
Majority	−0.6967	0.0396	0.498 (0.257–0.967)
Always	−0.4864	0.0434	0.615 (0.383–0.986)
Sub-analysis of subjects in the random allocation sites			
Age *	0.0426	0.0009	1.044 (1.018–1.070)
Number of sexual partners in the last two weeks *	−0.4152	0.0005	0.660 (0.522–0.835)
Sub-analysis of subjects in the free-choice sites			
Medication regimen (daily PrEP group) *			
event-driven group	2.5891	<0.0001	13.317 (6.664–26.614)
Medicine makes me safe, away from AIDS (Completely none) *			
A little	0.3956	0.2731	1.485 (0.732–3.013)
Somewhat	1.7924	<0.0001	6.004 (2.725–13.231)
Majority	1.4685	0.0224	4.343 (1.231–15.313)
Always	2.2687	0.0582	9.667 (0.925–101.068)
I am worried that others know I am taking medicine *			
A little	1.1294	0.0045	3.094 (1.420–6.741)
Somewhat	1.3899	0.0053	4.014 (1.510–10.672)
Majority	−0.1518	0.7839	0.859 (0.290–2.543)
Always	0.2005	0.5757	1.222 (0.606–2.466)

Note: * indicates missing data.

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
