# Peer review of "Study on Pre-Exposure Prophylaxis Regimens among Men Who Have Sex with Men: A Prospective Cohort Study"

_ijerph, 2019, doi:10.3390/ijerph16244996_

Round 1
Reviewer 1 Report
Abstract
Page 1, Line 29, remove sexual behavior, it is also behavioral characteristics. Page 1, Line 37, incidence rate are still high in both daily prep and event driven prep group, are they significantly lower than the control group. Please also report the incidence rate in control group. It is only meaningful to compare the effectiveness of these two regimen when they are both effective as compare to the blank control. Page 1, Line 42-43, specify the directions of the association, which are the risk factors, which are the protective factors.
Introduction
There are many typos and grammar errors, need language editing throughout the manuscript. Page 2, line 21, the author mentioned china cdc attempted to scale up prep in seven provinces, what were the results? and why it failed? What were the reasons for low awareness and willingness to use? In the introduction, the author should add some information about the importance of biomedical interventions such as prep to HIV prevention and control in Chinese MSM, in addition to behavioral intervention.
Method
Page 2, line 32, please clarify the background of NGOs, such as their scale, and what were their roles in this study, were they only responsible for part of the recruitment? Did they involved in the implementation and management of the project. Page 2, line 34, what do you mean public places? What are the proportion of the participants by four recruitment methods. Are the participants statistically different in the background characteristics by the four recruitment methods. Page 2, line 37, HIV antibody-negative, is it self-reported? Page 2, line 43, ……was designed as an open completely randomized….., but in the abstract, the author reported that this is a non-randomized study. Page 2, line 37-40, Usually, prep is only recommended to people at high-risk, how is this reflected in the inclusion criteria? Page 3, line 7, ….. 2 hours of sexual activity, change this to “2 hours after sexual activity”. Please clarify if the participants were managed by different research groups at different sites. Page 3, line 24, ……clinical and laboratory tests that the patients may have undergone in the recent two weeks. Please specify what tests were included in follow-ups? Are they the same as baseline? Page 3, line 27. How about the evaluation of serum TDF concentration in other two sites, i.e., Xinjiang and Guangxi. Page 3, line 37. ……average serum concentration of TDF……, please clarify if it is the average serum concentration of TDF of all serum samples collected for each patients during the full observation period. Page 3, line42-43. ……using cross-sectional baseline data and data obtained at first follow-up for each subject. Please justify the reason for only using data from first follow up instead of using average medication adherence rate during the full observation period. Page 4, line 21-23, “UNIVARIATE ANALYSIS……were analyzed” this sentence is not readable, the authors may need a native speaker to edit the language throughout the manuscript. Page 4, line 23-29, “the adherence level…….and efficacy cognition”, this part should be put in the measures section, and describe in detail such as the recall time for each variable.
Results
The authors used “subjects”, “participants”, and “patients”, please be consistent in one manuscript. Tables, the first column is quite hard to read, hard to separate the variable name and its category. Page 9, table 3. Please add comparison of serum TDF concentration within daily prep group by medication adherence rate. Also add comparison of serum TDF concentration within event-driven prep group by medication adherence rate. Page 9, line 22. “psychological characteristics”, I think these items are not psychological characteristics, they are more like “perception”. Please report average follow-up times. Page 12, line 30-31. “this may be because……was high”, I am not convinced with this statement. It is more likely that MSM with fewer partners perceive lower risk, thus they are less likely to adhere to prep.

Reviewer 2 Report
See attachment

Round 2
Reviewer 1 Report
just one minor error need to be corrected
Response to comment: Page 1, Line 42-43, specify the directions of the association, which are the risk factors, which are the protective factors
Response:
Thank you for your comments, we have revised the statement as ’ Subjects who were in the event-driven PrEP group (OR=2.152, 95% CI: 1.566–2.957), had fewer male sexual partners in the last two weeks (OR=0.685, 95% CI: 0.563–0.834), were one year older on average (OR=1.022, 95% CI: 1.002–1.043), considered that medication kept them safe, and were less worried about others knowing they took medicine, were more likely to have high adherence.’
"had fewer male sexual partners in the last two weeks (OR=0.685, 95% CI: 0.563–0.834)" this factor is a risk factor!
Reviewer 2 Report
Thank you for your detailed response.